# Fucoidan and Fucoxanthin Attenuate Hepatic Steatosis and Inflammation of NAFLD through Modulation of Leptin/Adiponectin Axis

**DOI:** 10.3390/md19030148

**Published:** 2021-03-12

**Authors:** Ping-Hsiao Shih, Sheng-Jie Shiue, Chun-Nan Chen, Sheng-Wei Cheng, Hsin-Yi Lin, Li-Wei Wu, Ming-Shun Wu

**Affiliations:** 1Center for Cell Therapy, Department of Medical Research, China Medical University Hospital, Taichung 404332, Taiwan; T31161@mail.cmuh.org.tw; 2Division of Gastroenterology, Department of Internal Medicine, Wan Fang Hospital, Taipei Medical University, Taipei 116, Taiwan; 108326@w.tmu.edu.tw (S.-J.S.); 86662@w.tmu.edu.tw (C.-N.C.); 97427@w.tmu.edu.tw (S.-W.C.); 3Department of Chemical Engineering and Biotechnology, National Taipei University of Technology, Taipei 10608, Taiwan; hbrunken@ntut.edu.tw; 4Department of Internal Medicine, National Taiwan University Hospital, YunLin Branch, YunLin 640, Taiwan; A00833@ntucc.gov.tw; 5International PhD Program in Medicine, College of Medicine, Taipei Medical University, Taipei 106, Taiwan; 6Division of Gastroenterology and Hepatology, Department of Internal Medicine, School of Medicine, College of Medicine, Taipei Medical University, Taipei 106, Taiwan; 7Integrative Therapy Center for Gastroenterologic Cancers, Wan Fang Hospital, Taipei Medical University, Taipei 106, Taiwan

**Keywords:** NAFLD, randomized controlled trial, lipid metabolism, lipotoxicity, liver fibrosis, adiponectin

## Abstract

Non-alcoholic fatty liver disease (NAFLD) is the emerging cause of chronic liver disease globally and lack of approved therapies. Here, we investigated the feasibility of combinatorial effects of low molecular weight fucoidan and high stability fucoxanthin (LMF-HSFx) as a therapeutic approach against NAFLD. We evaluated the inhibitory effects of LMF-HSFx or placebo in 42 NAFLD patients for 24 weeks and related mechanism in high fat diet (HFD) mice model and HepaRG^TM^ cell line. We found that LMF-HSFx reduces the relative values of alanine aminotransferase, aspartate aminotransferase, total cholesterol, triglyceride, fasting blood glucose and hemoglobin A1c in NAFLD patients. For lipid metabolism, LMF-HSFx reduces the scores of controlled attenuation parameter (CAP) and increases adiponectin and leptin expression. Interestingly, it reduces liver fibrosis in NAFLD patients, either. The proinflammatory cytokines interleukin (IL)-6 and interferon-γ are reduced in LMF-HSFx group. In HFD mice, LMF-HSFx attenuates hepatic lipotoxicity and modulates adipogenesis. Additionally, LMF-HSFx modulates SIRI-PGC-1 pathway in HepaRG cells under palmitic acid-induced lipotoxicity environment. Here, we describe that LMF-HSFx ameliorated hepatic steatosis, inflammation, fibrosis and insulin resistance in NAFLD patients. LMF-HSFx may modulate leptin-adiponectin axis in adipocytes and hepatocytes, then regulate lipid and glycogen metabolism, decrease insulin resistance and is against NAFLD.

## 1. Introduction

During the past century, dramatic modifications in lifestyle have radically changed the health priorities in most areas of the world, owing to a growing incidence of noncommunicable disease. The new epidemic in chronic liver disease is related to nonalcoholic fatty liver disease (NAFLD) and the estimated prevalence of NAFLD worldwide is approximately 25% [1]. NAFLD, particularly its histological phenotype non-alcoholic steatohepatitis (NASH), can potentially progress to advanced liver disease, fibrosis, cirrhosis and, ultimately, hepatocellular carcinoma [2]. NAFLD is considered to be the hepatic component of metabolic syndrome as its features are similar to those of metabolic disorders such as central obesity, inflammation, insulin resistance, hyperlipidemia, hyperglycemia, hypertension and type 2 diabetes [3]. Thus, it is important to treat NAFLD as well as its associated metabolic diseases. Given the lack of effective treatments for NAFLD, interventions targeting the hepatic fat accumulation, oxidative stress and inflammation are considered as promising therapeutic perspectives.

Fucoidan, kind of seaweed extracts, is a class of L-fucose-enriched sulfated polysaccharides and is extensively investigated for its different biological activities on anti- inflammatory, anti-aggregation and anti-oxidative effects [4,5,6,7]. The low molecular weight fucoidan (LMF) was reported to attenuate liver injury by inhibiting oxidative damage and inflammation [8]. Fucoxanthin, a carotenoid present in the chloroplasts of brown seaweeds, also possesses strong anti-inflammatory activity and cancer prevention through their antioxidant activities [9]. In addition, the anti-obesity function of high stability fucoxanthin (HSFx) was suggested and extensively discussed [9,10]. Herein, novel agents, like LMF and HSFx, with multiple functions, such as antioxidant, anti-inflammatory and anti-lipemic properties have great potential to NAFLD treatment. With focuses on the therapeutic effects of LMF plus HSFx for NAFLD, we investigated the clinical effects of LMF-HSFx for the insulin resistance and inflammation in NAFLD patients. For hepatic steatosis and fibrosis, we used non-invasive Fibroscan^®^ to evaluate the controlled attenuation parameter (CAP) and transient elastography, respectively [11,12]. We also examined the possible mechanisms of LMF-HSFx to against hepatic lipotoxicity in high fat diet (HFD) mice and HepaRG^TM^ cells.

## 2. Results

### 2.1. Low-Molecular Weight Fucoidan Plus High-Stability Fucoxanthin (LMF-HSFx) Attenuates Hepatic Lipotoxicity in Patients with NAFLD in Randomized Controlled Clinical Trial

Forty-two patients were equally randomized into LMF-HSFx and placebo group. The baseline characteristics of both groups were similar (Appendix A). There was no significant difference in anthropometry and laboratory data. Similar levels of stiffness (kPa, 6.5 ± 1.8 vs. 6.7 ± 2.9, *p* = 0.52), age (years, 55 ± 12.5 vs. 59 ± 10.5, *p* = 0.23) were found in two groups. The steatosis (CAP value) was worse in LMF-HSFx group (dB/m, 343.6 ± 44.6 vs. 303.6 ± 66.9, *p* = 0.03).

To examine the protective effect of LMF-HSFx on NAFLD-induced hepatic lipotoxicity, we investigated the serum biochemical parameters in NAFLD patients with *24*-weeks oral LMF-HSFx (825 mg LMF fucoidan plus 825 mg HSFx twice daily) or placebo (1650 mg cellulose powder twice daily) treatment (Figure 1). After 6 months, the relative ratio to baseline (%) of serum alanine aminotransferase (ALT) and aspartate aminotransferase (AST) were significantly decreased in LMF-HSFx group (Figure 2A,B, * *p* < 0.05, compared to placebo). The relative ratio of total cholesterol (TC) and triglyceride (TG) to baseline (%) were also significantly decreased at the end of 6 months (Figure 2C,D, * *p* < 0.05, compared to placebo). All the data indicate that hepatic lipotoxicity in NAFLD patients is attenuated by LMF-HSFx supplementation.

### 2.2. LMF-HSFx Reduces the Hepatic Steatosis and Fibrosis in Patients with NAFLD in Clinical Trial

After LMF-HSFx treatment for 6 months, the relative ratio of CAP value to baseline (%) was significantly decreased (Figure 3A, * *p* < 0.05, compared to placebo) in LMF-HSFx group. Interestingly, the relative ratio of stiffness degree to baseline (%) was also significantly decreased (Figure 3B, * *p* < 0.05, compared to placebo). All the data indicated that LMF-HSFx attenuates hepatic steatosis and fibrosis in NAFLD patients.

### 2.3. LMF-HSFx has the Potential to Attenuate the NAFLD-Induced Inflammation and Modulate Adipogenesis in Clinical Trial

The proinflammatory cytokines and adipokines are the key mediators in the pathogenesis of NAFLD [13]. Herein, we particularly focus on the changes of interleukin (IL)-6 and interferon (IFN)-γ among proinflammatory cytokines; and adiponectin and leptin among adipokines in NAFLD patients with LMF-HSFx or placebo treatment. Significant reductions were observed in IL-6 and IFN-γ levels at 3rd and 6th month in LMF-HSFx group (Figure 4A,B).

Adiponectin is an anti-diabetic adipokine, modulates lipid and glucose metabolism, including promotion of fatty acid oxidation and glucose utilization and repression of hepatic gluconeogenesis and decreased levels of adiponectin in obesity has been associated with mitochondrial dysfunction and insulin resistance in diabetes [14]. The other major adipokine is leptin, the net action of leptin is to inhibit appetite, stimulate thermogenesis, enhance fatty acid oxidation, decrease glucose and reduce body weight and fat [15]. For adipokines, both the relative ratios to baseline (%) of adiponectin and leptin were significantly increased (Figure 4C,D, * *p* < 0.05, compared to placebo).

### 2.4. LMF-HSFx has the Potential to Reduce the Insulin Resistance in Patients with NAFLD in Clinical Trial

Next, we investigated whether LMF-HSFx could attenuate insulin resistance in NAFLD. The data revealed that the level of fasting blood glucose (AC) and hemoglobin A1c (HbA1c) significantly decreased after 6 months LMF-HSFx treatment (Figure 5A,B, * *p* < 0.05, compared to placebo). The relative ratio to baseline (%) of insulin level was increased in LMF-HSFx group (Figure 5C). We further calculated the Homeostatic Model Assessment of Insulin Resistance (HOMA-IR) and insulin secretion index as the indicators of insulin resistance and beta cell function [16]. Interestingly, the lower HOMA-IR (Figure 5D) but higher insulin secretion index (Figure 5E) in the LMF-HSFx group compared to the placebo group, suggesting LMF-HSFx has the potential to attenuate the insulin resistance and to enhance the beta cell repair in NAFLD. Similar results were also observed in pre-diabetes mellitus (preDM) patients (Appendix A).

### 2.5. LMF-HSFx Attenuates Hepatic Lipotoxicity and Modulates Adipogenesis in Mice Fed with High-Fat Diet

Next, we investigated the effect of LMF-HSFx on hepatic lipotoxicity in mice. Figure 6A,B show the histochemistry stain and lipid droplet size of liver from HFD-fed mice treated with or without orally gavaged LMF-HSFx (400 mg/kg/BW/day) for 16 weeks. HE staining of liver tissue showed that HFD obviously induced abundant lipid droplet accumulation accompanied with ballooning and necrotic hepatocyte (Figure 6A, upper right panel), but LMF-HSFx significantly inhibited the lipid accumulation in liver (Figure 6A, lower panels). Neither control (normal diet) nor LMF-HSFx-treated group (normal diet) show any fatty liver phenomenon (Figure 3A, upper left and central panels, respectively). We further analyzed the amount and the size of lipid droplet in liver tissue, the data suggested that supplementation of LMF-HSFx significantly decreased the volume of lipid droplet in liver of mice fed with HFD (Figure 6B). Figure 6C shows that the level of fasting blood glucose significantly increased in HFD mice compared with that in the ND mice, while 400 mg/kg/BW/day of LMF-HSFx treatment significantly (*p* < 0.05) reduced serum glucose level in HFD mice. TG content in HFD mice with LMF-HSFx group was also lower than that in the HFD group (Figure 6D). Furthermore, the serum AST and ALT levels were significantly increased in HFD mice as compared with those in ND mice, suggesting LMF-HSFx significantly decreased the HFD-induced lipotoxicity in liver (Figure 6E,F). Following, we evaluated the effects of LMF-HSFx on HFD-induced adipogenesis dysregulation in mice. In adipose tissue and pro-brown adipose tissue, LMF-HSFx also significantly up-regulated the adiponectin expression genes, *adipoq* and *adig* and leptin expression gene *lep* (Table 1).

### 2.6. LMF-HSFx Modulates PGC-1-Medicated Pathways in Palmitic Acid-Treated HepaRG Hepatocytes

In HepaRG cells, the PA-induced hepatic lipotoxicity resulted in the decrease of SIRT family protein expression (Figure 7A,B), which are modulated by leptin- adiponectin axis [17]. In addition, we found that PA treatment notably induced cleavage or decrease in SIRTs expression in a dose-dependent manner (cells without PA, with PA200, with PA400), while LMF-HSFx significantly restored PA-induced SIRT2, 3 and 6 degradation (Figure 7A). Interestingly, the levels of SIRT1-3, 6 were significantly enhanced in the LMF-HSFx group as compared with control group (Figure 7B). Furthermore, PGC-1α was increased but PGC-1β was decreased in PA-treated hepatocytes in a dose-dependent manner (cells without PA, with PA200, with PA400), while LMF-HSFx significantly restored the effects (Figure 7C,D). Coordinately, the level of down-stream lipid metabolism-related protein, adipose triglyceride lipase (ATGL), was also significantly restored by the treatment of LMF-HSFx 25 μg/mL (Figure 7C,D).

### 2.7. LMF-HSFx Ameliorates PA-Induced Early Apoptosis and Cellular Mitochondrial Disruption in HepaRG Cells

The protect effects of LMF-HSFx on cell apoptosis in PA-treated HepaRG cells was investigated. Figure 8A,B was revealed that 200 and 400 μM of PA significantly (*p* < 0.05) increased the proportion of cells in early apoptosis (Annexin V + /PI− cells; 11 ± 2.7 and 29 ± 5.8%, respectively) compared with the control group (3 ± 1.8%). LMF-HSFx restored PA-induced cell death and DNA fragmentation of HepaRG cells (Appendix A). Also, LMF-HSFx restored the PA-induced cell cycle arrest (Appendix A) and reduced the PA-induced Caspase 3 activation in HepaRG cells (Appendix A). We further investigated the ΔΨm in PA-treated HepaRG cells with or without LMF-HSFx 25 μg/mL. In the JC-1 staining assay, it was identified that PA induced decrease of mitochondrial potential is a dose-dependent manner (Figure 8C,D). LMF-HSFx induced a significant increase in mitochondrial integrity as compared with the control group (Figure 8C,D).

## 3. Discussion

NAFLD is excessive fat build-up in the liver without another clear cause such as alcohol use. Hepatic lipid accumulation results from an imbalance between lipid availability and lipid disposal. These toxic lipid species induce hepatocellular stress, injury and death, leading to fibrogenesis and genomic instability. Here, in randomized controlled clinical trial, we provide evidence that LMF-HSFx has the potential against lipotoxicity (Figure 2), hepatic steatosis (Figure 3), inflammation (Figure 4) and insulin resistance (Figure 5) in NAFLD patients. In addition, LMF-HSFx enhancing the leptin and adiponectin expression in NAFLD patients suggested that LMF-HSFx may modulate adipogenesis in NAFLD patients. Similar results were also confirmed in HFD-induced-NAFLD mice. LMF-HSFx also significantly decreased the HFD-induced hepatic lipotoxicity in mice, such as lipid droplet sizes, blood glucose level, triglyceride level, AST and ALT (Figure 6). Following, LMF-HSFx also significantly up-regulated the adiponectin expression genes, *adipoq* and *adig* and leptin expression gene *lep* in adipose tissue and pro-brown adipose tissue in HFD-fed mice (Table 1). Briefly, in vivo, LMF-HSFx may enhance the leptin and adiponectin expression in adipocytes and decreases insulin resistance. Next, adiponectin triggers the Adiponectin-AdipoR1/2 and SIRI-PGC-1 pathways [17,18,19] in the hepatocyte cell (Figure 7) and decrease the liver lipid disposal (Figure 6). Since the Adiponectin-AdipoR1/2-SIRI-PGC-1 pathways play crucial roles for mitochondrial function, gluconeogenesis, fatty acid oxidation and De novo lipogenesis in liver, we suggest that LMF-HSFx affects lipid and glycogen metabolism, mitochondrial function through Adiponectin-AdipoR1/2 and SIRI-PGC-1 pathways in adipocytes and hepatocytes (Figure 2, Figure 6, Figure 7 and Figure 8 and Table 1), alleviates insulin resistance (Figure 5 and Appendix A), inflammation and hepatic steatosis (Figure 3 and Figure 4 and Appendix A). Based on the mechanisms, LMF-HSFx sheds light on treatment for NAFLD (Figure 9).

LMF-HSFx suppresses lipotoxicity-induced liver injury. SIRI-PGC-1 pathways are recognized as interesting targets for NAFLD because they are involved in lipid metabolism, mitochondrial activation and inflammation in liver [20,21]. PGC-1α is a downstream sensor of metabolic, hormonal and inflammatory signals that is responsible for the balance of hepatic gluconeogenesis, fatty acid β-oxidation and mitochondrial biogenesis [22]. In this study, LMF-HSFx reduced the lipotoxicity-induced PGC-1α over expression (Figure 7) suggesting LMF-HSFx could repress the lipotoxicity-induced gluconeogenesis. Functionally, PGC1β is a powerful inducer of mitochondrial biogenesis and respiration [23]. It is also involved in the expression of the lipogenic programmer in liver [24] and plays a central role in interferon-γ-induced host defense [25]. In this study, LMF-HSFx significantly restored lipotoxicity-induced PGC-1β decrease. Coordinately, the level of down-stream lipid metabolism-related protein, ATGL, was also significantly restored by LMF-HSFx. All the data suggest that LMF-HSFx modulates lipolysis pathways, gluconeogenesis and mitochondrial function in liver, especially the SIRI-PGC-1 axis pathways. Similar pathway modulations by fucoidan or fucoxanthin alone were reported in previous animal studies. Fucoidan prevents NAFLD through SIRI-PGC-1 pathway in db/db mice [26]. Fucoxanthin affects the lipid metabolism through the leptin and adiponectin-mediated pathways for anti-obesity [10].

The pharmacokinetic and tissue distribution of active compounds is essential for drug development process. In rats, fucoidan favorably accumulates in kidneys, spleen and liver [27]. Interestingly, the blood residence time of fucoidan is about 7 h suggesting it shows long absorption time and extended circulation in the blood [27]. In mice, fucoxanthin preferentially accumulates in the heart, liver and adipose tissue [28]. The time at maximum concentration (Tmax) of fucoxanthin in adipose tissue was 24 h, while the Tmax in the others was 4 h [28]. Therefore, in Hwang et al. study, LMF was more effective than HSFx at improving hepatic glucose metabolism in the diabetic mice. As for adipose tissue, glucose and lipid metabolism was significantly upregulated by HSFx and LMF-HSFx [29]. In Hwang et al. study, the decrease in urinary sugar and reduce in inflammatory adipocytokines were significantly observed in LMF-HSFx group, but not in HSFx or LMF alone [29]. Interestingly, the regulation efficacies of LMF-HSFx were better than HSFx or LMF alone on urinary sugar decreasing, glucose and lipid metabolism in white adipose tissue, indicating a synergistic effect of LMF and HSFx [29].

LMF-HSFx suppresses the liver fibrosis. After LMF-HSFx treatment for 6 months, interestingly, the relative ratio to baseline (%) of stiffness degree was also significantly decreased (Figure 3B, * *p* < 0.05, compared to placebo). Moreover, we analyzed the stiffness degree change of 14 patients (8 people in LMF-HSFx group and 5 in placebo group) with mild-severe stiffness or cirrhosis during treatment period. In LMF-HSFx group, there were 4 patients whose stiffness degree decreased down to less than F1, that is, no fibrosis at 6th month. In contrast, no one in placebo group recovered from mild -severe stiffness or cirrhosis (Appendix A). In rat, fucoidan alone was reported to suppress CCl_4_-induced hepatic fibrosis through protection of hepatocytes and inhibition of hepatic stellate cell proliferation [30]. Similar effects were observed for Fucoxanthin alone. In choline-deficient L-amino acid-defined high fat diet -induced NASH model mice, fucoxanthin restrained not only hepatic oxidative stress and inflammation but also early phase of fibrosis [31]. Proinflammtory cytokines, such as IL-6 play an important role in liver fibrosis. Previous study reported that catalase, anti-IL-6, or siRNA-IL-6 inhibited the phosphorylation of p38 in Kupffer cells which co-cultured hepatic stellate cell and also blocked TIMP1 upregulation and collagen I accumulation for liver fibrosis [32]. In this study, LMF-HSFx induced-IL-6 and IFN-γ suppression may play a role to reverse liver fibrosis in NAFLD patients. Further analysis of the effects of LMF-HSFx on the correlation of IL-6 and Kupffer cells are required to reveal this mechanism.

LMF-HSFx suppresses the insulin resistance. Insulin resistance is a common feature of NAFLD that contributes to its pathogenesis [33]. Insulin resistance is characterized by reduced glucose disposal in non-hepatic tissues, including adipose tissue and muscle [34]. Studies have shown metabolic crosstalk between adipose tissue and the liver. Adiponectin released by adipose tissue has protective effects in the liver; but IL-6 released by adipose tissue has proinflammtory effects and enhances the NAFLD development in liver [35]. In this study, LMF-HSFx enhances the Adiponectin expression but suppresses the IL-6 level in NAFLD patients (Figure 4A,C). In addition, LMF-HSFx modulates adipogenesis in HFD mice adipose tissue (Table 1). All the data suggest LMF-HSFx may reduce the insulin resistance in NAFLD patients through the regulation of Adiponectin-AdipoR1/2 pathways [36], which contribute the critical part of insulin resistance. Similar pathway modulation for insulin resistance by fucoidan or fucoxanthin alone was reported in previous NAFLD animal model studies [10,37]. In this study, LMF-HSFx also has the potential to promote beta cell function in NAFLD patients. LMF-HSFx reduced the relative ratio of AC and HbA1c after 6 months treatment. Interestingly, LMF-HSFx enhance insulin secretion and the beta cell function after 6 months treatment (Figure 5). Similar results were also observed in preDM patients (Appendix A).

There are several clinical implications of our findings. First, LMF-HSFx attenuates the hepatic steatosis and fibrosis in NAFLD patients (Figure 3). Second, LMF-HSFx has the potential to reduce the insulin resistance and regulates glycogen metabolism of hepatocytes and beta cells in NAFLD patients. In this report, LMF-HSFx showed significantly glycemic homeostasis with reduction of HbA1c and narrow standard error (Figure 5B). In addition, the preDM patients exhibited more stable HOMA ratio in the LMF-HSFx group (Appendix A). To our knowledge, this is the first study investigating the glycemic control of LMF-HSFx on NAFLD. Third, Fucoidan, a kind of prebiotics, may regulate the gut microbiota in NAFLD. Modulation of the gut microbiota represents a new treatment for NAFLD [38]. There are numerous animal and human studies that point to the potential of prebiotics to treat NAFLD [39]. Therefore, a more extensive investigation is required.

In conclusion, we have uncovered a new approach to ameliorate NAFLD since LMF-HSFx is able to against lipotoxicity, inflammation, insulin resistance and hepatic steatosis. In addition, our data substantiate the hypothesis that NAFLD-induced liver damage is associated with an abnormal SIRT-PGC-1 signaling; however, all these changes were notably reversed by LMF-HSFx treatment. In addition, combination of LMF and HSFx also shows a synergistic modulation on glucose and lipid metabolism [29]. This report demonstrated for the first time that LMF-HSFx could attenuate hepatitis and even liver fibrosis in NAFLD patients. However, relative shorter period of observation may limit our understanding with respect to the long-term application. In addition, more precise conclusion can be achieved by increasing the sample size and including liver biopsies. Future studies should increase the number of patients and include long-term follow-up assessment.

## 4. Materials and Methods

### 4.1. Materials

The LMF-HSFx capsules in this trial were derived from *Sargassum hemiphyllum* and prepared by HiQ Marine Biotech, Taipei, Taiwan. The average molecular weight, monosaccharide fucose content and sulfate content of LMF used in this study, were 0.8 KDa (92.1%), 210.9 ± 3.3 mmol/g and 38.9% ± 0.4% (*w*/*w*), respectively [40]. HSFx is a mixture containing about 10% of fucoxanthin that is coated directly with polysaccharides of its own [41].

### 4.2. Clinical Trial Design

The study was conducted in the Taipei Medical University-Wanfang Hospital in Taiwan. The screening period was between Dec. 2016 to Feb. 2017, seventy patients who visited the outpatient department of Gastroenterology and Hepatology, age ranging from 20 to 75 years old, were screened with sonographic evidence of fatty liver. The criteria of exclusion were pregnancy, short life expectancy (<6 months), medication of anti-diabetic agent such as Liraglutide or Pioglitazone, supplement with Vitamin E, viral hepatitis, autoimmune disease, allergy to seafood and people who decline to participate or sign the informed consents (Figure 1). After explanation of informed consent and repeated confirmation of fatty liver by FibroScan^®^ and CAP, forty-two patients, age (years, 55 ± 12.5 vs. 59 ± 10.5, *p* = 0.23), were enrolled into the study and randomized into two groups, the treatment and the placebo group. Subjects took 3 capsules of LMF-HSFx (each capsule contains 275 mg LMF and 275 mg HSFx, Hi-Q Marine Biotech International Ltd. Taipei, Taiwan) twice per day in the treatment group, or placebo (3 capsules of 550 mg/capsule cellulose powder, Sigma-Aldrich, St. Louis, MO, USA) in the control group. Both groups were implemented with a twenty-four-week follow-up period. The severity of liver steatosis, fibrosis, AST, ALT, TC, TG, Creatine and fasting blood sugar were monitored every four weeks during the follow-up period. Metabolic profiles such as adiponectin and leptin were measured at the baseline, 4th, 12th and 24th week during follow-up period. The protocols were approved by the Joint committee of Institutional review board (J-IRB) of Taipei Medical University and then registered at ClinicalTrials.gov (NCT02875392) (accessed on 11 March 2021). Non-invasive evaluation of liver steatosis and fibrosis in patients with suspected NAFLD by Fibroscan^®^ vibration-controlled transient elastography CAP and liver stiffness measurement is routinely undertaken in clinical practice [42]. Ranksum test was applied for the statistic comparison of the baseline characteristics between treatment and placebo groups. Statistically significant were achieved when the *p* value is less than 0.05 for the tests. The LMF-HSFx capsules in this trial were derived from *Sargassum hemiphyllum* and prepared by HiQ Marine Biotech, Taipei, Taiwan. The average molecular weight, monosaccharide fucose content and sulfate content of LMF used in this study, were 0.8 KDa (92.1%), 210.9 ± 3.3 mmol/g and 38.9% ± 0.4% (*w*/*w*), respectively [40]. HSFx is a mixture containing about 10% of fucoxanthin that is coated directly with polysaccharides of its own [41].

### 4.3. Animal Design

All animal experiments were approved by the Taipei Medical University Committee of Experimental Animal Care and Use (approval No. LAC-2014-0414) and performed in accordance with relevant guidelines and regulations. Induction of NAFLD by HFD in C57BL/6 nude mice is used in this study [43]. Male 6-week-old mice were purchased from Bio LASCO Taiwan Co and housed at Animal Center of Taipei Medical University with temperature maintained at 22 ± 1 °C and with a 12 h light/12 h dark cycle. After one week, the mice were randomly divided into 6 groups (*n* = 6–8 per group, 3–4 mice per cage), normal diet (ND): AIN-93G purified rodent diet (200 g of casein and 70 g of soybean oil/kg diet) [44]; high-fat diet (HFD): AIN-93G purified rodent diet with 60% fat-derived calorie diet (Lasco, Taiwan) for sixteen weeks; ND or HFD plus orally gavaged with 200 or 400 mg/kg/BW/day of LMF-HSFx, respectively. Weight and food consumption were recorded every day and blood glucose level was examined from tail vein every week by biochemical analyzer. Blood collection was executed by retro-orbital sampling on the week fourth and on the end of study. Finally, the mice were fasting overnight and then anesthetized by inhalant anesthesia with isoflurane before sacrifice. For the serum biochemistry study, the blood was analyzed from each mouse by a biochemical analyzer to detect liver function and serum glucose. The right lobe of liver was fixed with 4% buffered paraformaldehyde and embedded in paraffin for further hematoxylin and eosin stain and left lobe of liver was soaked in liquid nitrogen and then frozen at −80 °C for further study.

### 4.4. Adipogenesis RT^2^ Profiler PCR Array

The HFD mice with or without 400 mg/kg/BW/day of LMF-HSFx through oral gavage. After 16 weeks treatment, the mice were anesthetized by inhalant anesthesia with isoflurane before sacrifice. RNA was extracted from adipose tissues by using a Direct-zol RNA Miniprep Plus kit (Cat. No. R2073, Zymo Research, Irvine, CA, USA) and its quantity and quality were estimated by NanoDrop measurement (ThermoFisher, Waltham, MA, USA). The RT PCR processing was performed using a commercially available RT^2^ Profiler™ PCR Array Mouse Adipogenesis (Cat. no. PAMM-049ZA, QIAGEN, Hilden, Germany) for genes of interest. Following, the fold change of target gene shows the ratio of treated group/control group and values > 1.5 or < 0.5 were considered significantly modified.

### 4.5. Cell Culture

Human HepaRG^TM^ cells (Lonza, Alpharetta, GA, USA) were cultured with basal medium supplemented with HepaRG^TM^ Maintenance and Metabolism Medium Supplement (Lonza). At the day of treatment, the medium was refreshed by HepaRG Induction Medium (Lonza) containing indicated concentrations of bovine serum-conjugated palmitic acid (PA) (Merck KGaA, Darmstadt, Germany) in the presence or absence of 25 μg/mL LMF-HSFx (HiQ Marine Biotech, Taipei, Taiwan).

### 4.6. Annexin V/Propidium Iodide (PI) Double Staining Assay

The HepaRG cells were cultured in control, LMF-HSFx 25 μg/mL, PA 200 or 400 μM pluswith or without LMF-HSFx 25 μg/mL for 24 h in HepaRG Induction Medium. At the end of the incubation, the floating and the adherent cells were harvested and then incubated with Annexin V- fluorescein isothiocyanate (FITC) and PI for 15 min at 37 °C in the dark after washing with cold PBS. Finally, the cells were re-suspended in 500 μL PBS and then analyzed by a flow cytometry (FACSCantoTM II, BD Biosciences, San Jose, CA, USA). A positive control group was established by exposing the cells to ultraviolet radiation half an hour for adjusting for fluorescence spectral overlap compensation. The data for 10,000 cells were collected and evaluated with BD FACSDiva software (v8.0- Becton, Dickinson and Company, Franklin Lakes, NJ, USA).

### 4.7. Mitochondrial Membrane Potential (ΔΨm) Evaluation

The HepaRG cells were cultured in control, LMF-HSFx 25 μg/mL, PA 200 or 400 μM pluswith or without LMF-HSFx 25 μg/mL for 18 h in HepaRG Induction Medium. At the end of the incubation, the floating and the adherent cells were harvested, washed with cold PBS and then incubated with 500 μL of JC-1 Working Buffer for 15 min at 37 °C in the dark after. Finally, the cells were re-suspended in 500 μL of Assay Buffer and then analyzed in the FL-1 (JC-1 Green) and FL-2 (JC-1 Red) channels by the flow cytometry. Data represented as the percentage of lowered red fluorescence + cells which is indicative of apoptosis were collected from 10,000 total cells and evaluated with BD Diva software.

### 4.8. Western Blot Analysis

The HepRG cells were cultured in control, LMF-HSFx 25 μg/mL, PA 200 or 400 μM with or without LMF-HSFx 25 μg/mL for 24 h in HepaRG Induction Medium. After treatment, the HepaRG cells were harvested and lysed in cold radioimmune precipitation assay (RIPA) buffer containing protease and phosphatase inhibitor cocktails (Merck KGaA, Darmstadt, Germany). Protein concentrations were measured by Bradford protein assay. Western blot processing was performed as previously reported [45]. Primary antibodies: SIRT-1, 2, 3, 4, 6, PGC-1α, PGC-1β, ATGL and control protein GAPDH (Millipore). Images were captured and analyzed by ChemiDoc™ Imaging System (BIO-RAD, Hercules, CA, USA).

### 4.9. Statistical Analysis

All assays were performed as three independent experiments with *n* ≥ 3 in each test. Values are expressed as the mean ± standard error of the mean (SEM). Statistically significant differences between treatment group and control group were assessed by one-way analysis (animal and cell culture studies) or two-way analysis (clinical trial) of variance followed by Tukey’s multiple comparison test using Prism 6.1d (GraphPad Software, Inc., La Jolla, CA, USA) and differences were considered statistically significant with probability (*p*) < 0.05.

## Figures and Tables

**Figure 1 marinedrugs-19-00148-f001:**
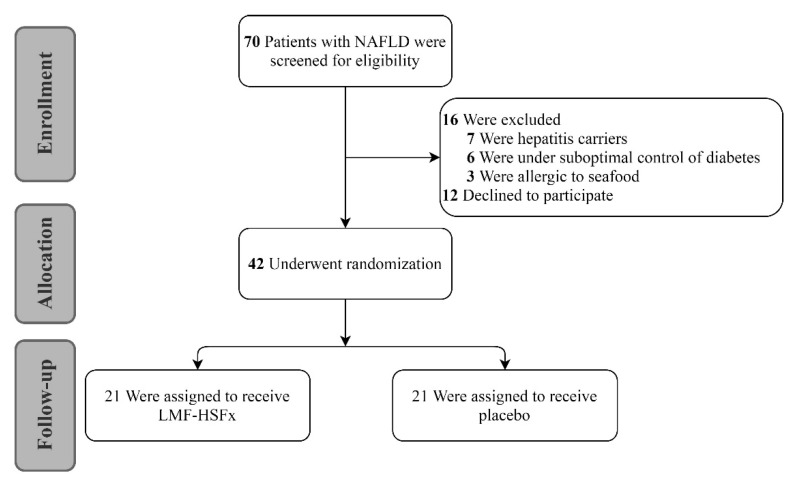
The consort diagram. Seventy patients were enrolled into the initial evaluation of the study. Sixteen patients were excluded owing to the possible bias from autoimmune hepatitis, seafood allergy and diabetes medication and twelve patients declined to participate. Forty-two patients were randomized into low-molecular weight fucoidan and high-stability fucoxanthin (LMF-HSFx) and placebo group.

**Figure 2 marinedrugs-19-00148-f002:**
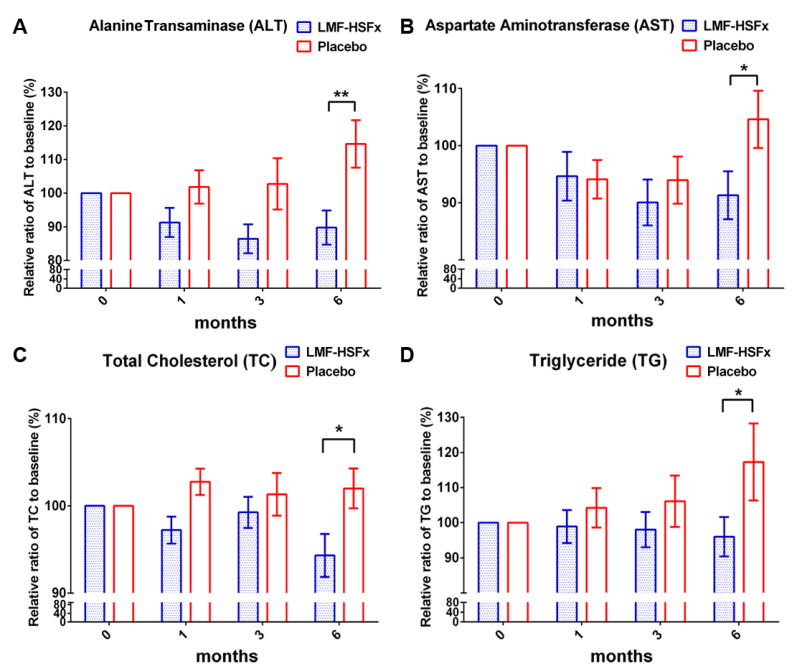
LMF-HSFx attenuates hepatic lipotoxicity in patients with nonalcoholic fatty liver disease. The graph demonstrated the change from baseline of (**A**) Alanine transaminase, ALT; (**B**) Aspartate aminotransferase, AST; (**C**) total cholesterol, TC; (**D**) triglyceride, TG; Significant reduction of ALT, AST, TC and TG were observed at 6th month in LMF-HSFx group (* *p* < 0.05, ** *p* < 0.01, compared with placebo).

**Figure 3 marinedrugs-19-00148-f003:**
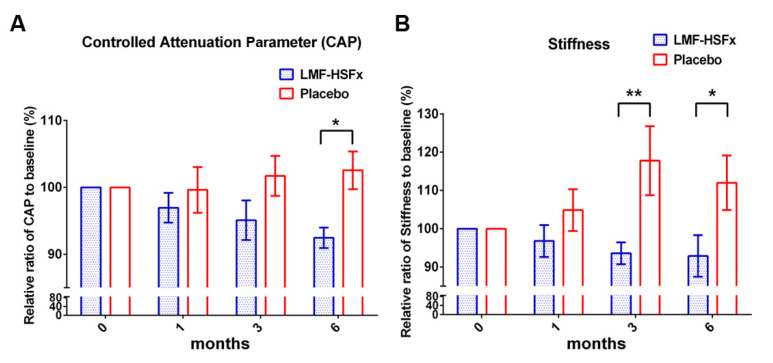
LMF-HSFx attenuates hepatic steatosis and fibrosis in patients with NAFLD. The graph demonstrated the change from baseline of (**A**) controlled attenuation parameter, CAP; (**B**) stiffness of patients with LMF-HSFx or placebo treatment. Significant reduction of CAP was observed at 6th month and significant reduction of stiffness was observed at 3rd and 6th month in LMF-HSFx group (* *p* < 0.05, ** *p* < 0.01, compared with placebo).

**Figure 4 marinedrugs-19-00148-f004:**
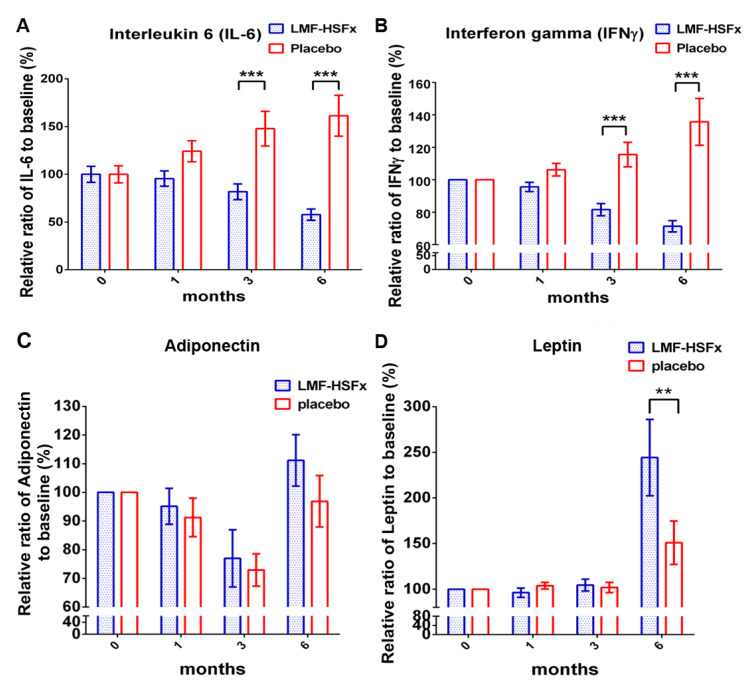
LMF-HSFx attenuates the NAFLD-induced inflammation and modulate adipogenesis. The graph demonstrated the change from baseline of serum (A) IL-6, (**B**) IFN-γ, (**C**) adiponectin and (**D**) leptin of patients with LMF-HSFx or placebo treatment. Significant reduction of IL-6 and IFN-γ change was observed at 3rd and 6th month in LMF-HSFx group (*** *p* < 0.001, compared with placebo). The significant increasing of leptin was observed at 6th month in LMF-HSFx group (** *p* < 0.01, compared with placebo).

**Figure 5 marinedrugs-19-00148-f005:**
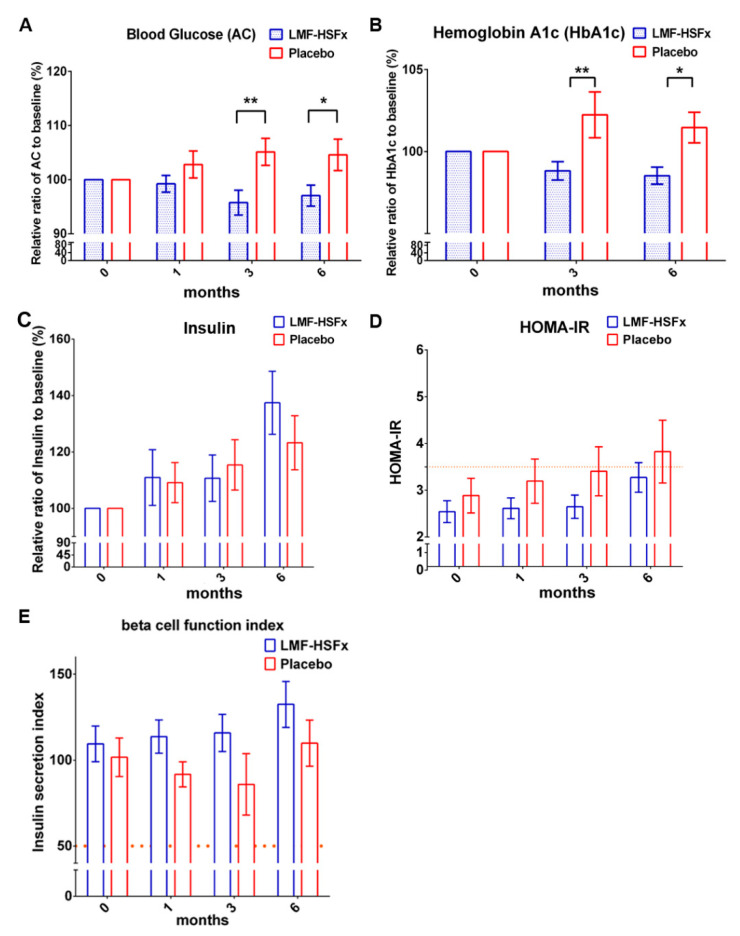
LMF-HSFx reduces the insulin resistance in patients with NAFLD. The graph demonstrated the change from baseline of (**A**) AC, (**B**) HbA1c, (**C**) insulin, (**D**) HOMA-IR and (**E**) the insulin secretion index (beta cell function index) of patients with LMF-HSFx or placebo treatment. Significant reduction of AC, HbA1c change was observed at 3rd and 6th month in LMF-HSFx group (* *p* < 0.05, ** *p* < 0.01, compared with placebo). (**D**) The average HOMA-IR of placebo not LMF-HSFx group was higher than 3.5 at 6th month. (**E**) The increasing of beta cell function index in LMF-HSFx group during the 6 months.

**Figure 6 marinedrugs-19-00148-f006:**
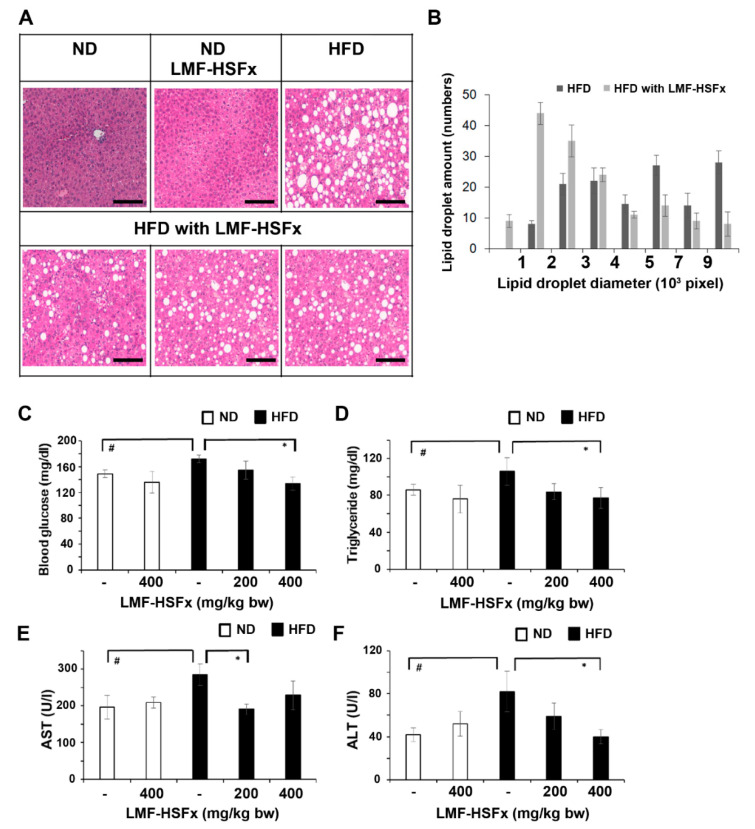
LMF-HSFx inhibits hepatic lipotoxicity in liver tissues of mice fed with high-fat diet. (**A**) Liver sections stained with Hematoxylin-eosin (HE) from high-fat diet (HFD) mice or normal diet (ND) mice with or without LMF-HSFx treatment. The HE staining of liver sections was performed at 16th week from ND mice or HFD mice with or without orally gavaged LMF-HSFx (400 mg/kg/BW/day). Scale bar: 100 μm. (**B**) LMF-HSFx decreases the volume of lipid droplet in HFD mice liver. The differential distribution of lipid droplets in HE stained-liver tissues of HFD mice with or without 400 mg/kg/BW/day LMF-HSFx treatment after 16 weeks. The *X*-axis is the lipid droplets diameter and *Y*-axis is the total lipid droplet numbers in 10 HE stained-liver images from one mouse, *n* = 3 mice. *X*-axis unit: 10^3^ pixel and *Y*-axis unit: the total lipid droplet numbers in 10 images of each area 2560 × 1922 pixel^2^. (**C**) The blood glucose (mg/dL, dL: 100 mL), (**D**) triglyceride (mg/dL), (**E**) AST (U/L) and (**F**) ALT (U/L) concertation in ND or HFD mice with or without 200 or 400 mg/kg/BW/day LMF-HSFx treatment through oral gavage after 16 weeks (^#^
*p* < 0.05, compared with DN mice; * *p* < 0.05, compared with HFD mice).

**Figure 7 marinedrugs-19-00148-f007:**
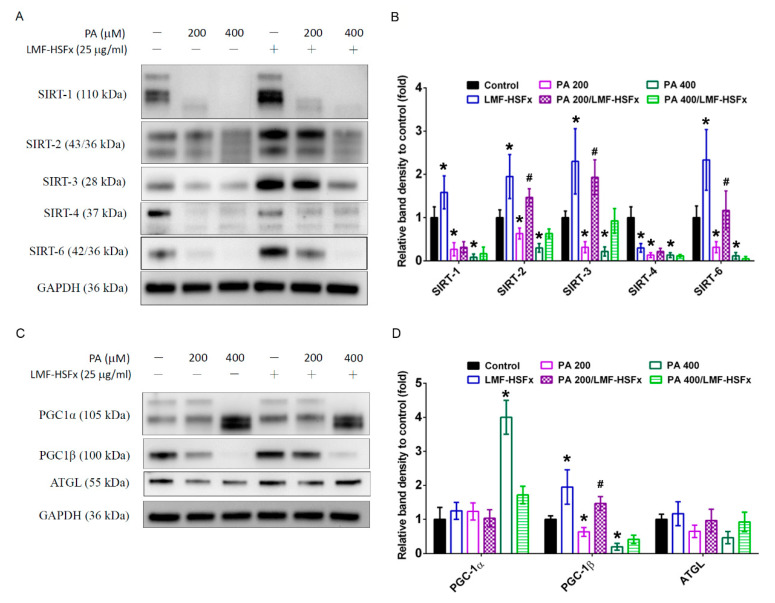
LMF-HSFx modulates SIRT-PGC-1 axis in palmitic acid-treated HepaRG hepatocytes. LMF-HSFx restored palmitic acid-induced SIRT2, 3, 6, PGC-1β and ATGL degradation. The HepaRG cells were cultured in PA 200 or 400 μM with or without LMF-HSFx 25 μg/mL. Representative Western blots of (**A**) SIRT-1, 2, 3, 4, 6, control protein GAPDH; (**C**) PGC-1α, PGC-1β, ATGL and GAPDH. Summarized bar graphs depicting the protein level of (**B**) SIRT-1, 2, 3, 4, 6; (**D**) PGC-1α, PGC-1β, ATGL. Each column represents mean ± SEM, taking the control group as 100% (* *p* < 0.05 compared with control; ^#^
*p* < 0.05, compared with PA200 group).

**Figure 8 marinedrugs-19-00148-f008:**
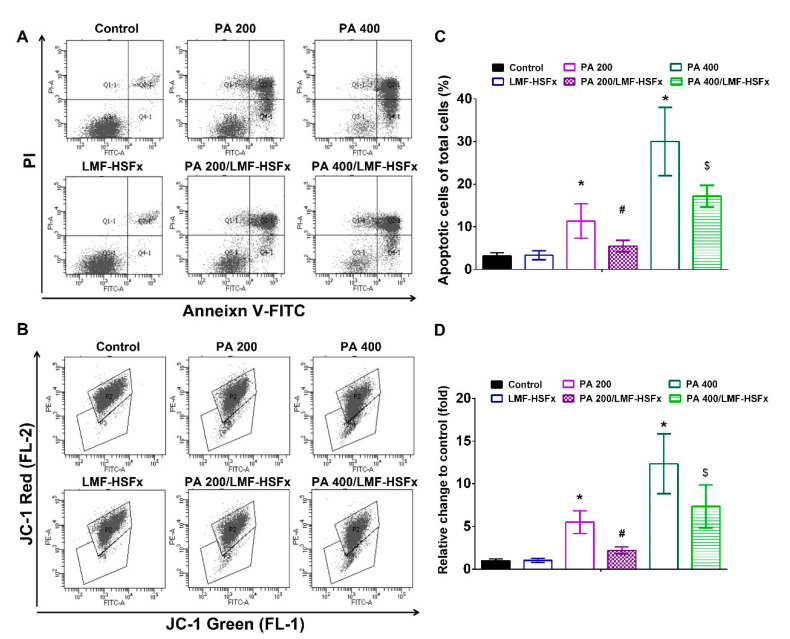
LMF-HSFx ameliorates PA-induced early apoptosis and cellular mitochondrial disruption in HepaRG cells. (**A**) the early apoptosis by Annexin V/PI stain and (**B**) mitochondrial disruption apoptosis by JC-1 labeling on flow cytometer of the HepRG cells in PA 200 or 400 μM with or without LMF-HSFx 25 μg/mL, LMF-HSFx 25 μg/mL and untreated control groups. (**C**) summarized bars depicting the differentiation percentage of apoptotic cells in groups by Annexin V/PI stain, taking total cells as 100%. (**D**) summarized bars depicting the Green/red fluorescence ratio in groups by JC-1 labeling, taking control group cells as 1fold (* *p* < 0.05, compared with control; ^#^
*p* < 0.05, compared with PA200 group; ^$^
*p* < 0.05, compared with PA400 group).

**Figure 9 marinedrugs-19-00148-f009:**
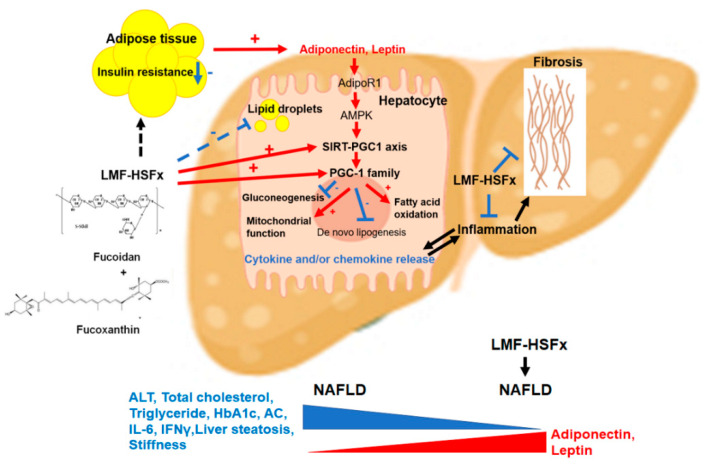
The work mechanism of LMF-HSFx for NAFLD. LMF-HSFx targets Adipocytes and hepatocytes. In adipose tissue, LMF-HSFx decreases the insulin resistance and enhances the Adiponectin and Leptin expression. In hepatocyte, LMF-HSFx directly activates SIRT-PGC1 axis and PGC-1 family expression. Combination with the stimulation effects from Adiponectin and Leptin, the SIRT-PGC1 axis mediated mitochondrial function and fatty acid oxidation are activated by LMF-HSFx but gluconeogenesis and De novo lipogenesis are inhibited by LMF-HSFx. LMF-HSFx also decreases the pro-inflammatory cytokine (IL6 and INFγ) release from hepatocyte and suppresses the fibrosis in NAFLD.

**Table 1 marinedrugs-19-00148-t001:** Effect of LMF-HSFx supplementation on high fat diet-induced adipogenesis dysregulation in mice.

Adipogenesis Regulation							
Total Adipose Tissue				In White Adipose Tissue Only		In Pro-Brown Adipose Tissue Only	
Gene	Fold *	Gene	Fold *	Gene	Fold *	Gene	Fold *
*Adig*	4.2	*Runx1t1*	14.9	*Gata2*	10.2	*Creb1*	16.8
*Adipoq*	13.8	*Shh*	5.8	*Klf3*	13.4	*Foxc2*	9.9
*Lep*	6.7	*Sirt1*	18.5			*Insr*	11.3
*Retn*	8.2	*Sirt2*	9.3			*Irs1*	12.9
*Agt*	39.2					*Ppara*	15.7
*Lipe*	6.6					*Ppard*	9.4
*Adrb2*	8.7						
*Lrp5*	11.3						
*Ncor2*	4.1						

* Fold change of target gene shows the ratio of HFD-LMF-HSFx/HFD. HFD-LMF-HSFx: HFD mice with 400 mg/kg/BW/day LMF-HSFx treatment through oral gavage after 16 weeks; HFD: mice with HFD after 16 weeks.

## Data Availability

Data is contained within the article or Appendix A.

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
