# Peer review of "Fucoidan and Fucoxanthin Attenuate Hepatic Steatosis and Inflammation of NAFLD through Modulation of Leptin/Adiponectin Axis"

_marinedrugs, 2021, doi:10.3390/md19030148_

Round 1

Reviewer 1 Report

The work by Ping-Hsiao Shih et al. describes the beneficial effects on NAFLD inflammation and hepatic function of Fucoidan and Fucoxanthin.

Although the work is clearly written and well-conceived, some modifications are required in order to render figure auto-explanatory.

  1. The legends should clarify all the abbreviations included in the figure (i.e. OF-HF).
  2. All the charts should be depicted in a legible size: charts in figure 7 are too much small
  3. Figure 7 and 8 charts x axis legend is too complicated: is better to use the same of the western blot images, that id simpler, moreover this will render more homogeneous the entire figure.

More specific issue could be raised for figure 3. This figure is not extensively commented in the text for instance stating differences between data for normal diet and for HFD. Moreover the charts are not clear because negative control is located at the center of values (I would suggest to reconsider output format of this charts). Moreover in H&E images should scale bar.

Generally all abbreviations should be explained in the text the first time of mention. Moreover a phrase resuming the treatment and the model (patients, mice or cells) used in the experiment at the beginning of each result session will render more clear all the paper.

Moreover no mention on Palmitic acid treatment was done  in Material and methods

Author Response

Response to Reviewer 1 comment

Point 1. The legends should clarify all the abbreviations included in the figure (i.e. OF-HF).

Response 1. Thank you for pointing this out. As you suggested to check all abbreviations in manuscript and figures and proofread them. “ OF-HF” in Figure 6, we now rewrite the abbreviation as “LMF-HSFx (mg/kg bw)”

Point 2. All the charts should be depicted in a legible size: charts in figure 7 are too much small.

Response 2. Thank you for pointing this out. As you suggested, we redraw the Figure 6-8, All the charts are rewrite and depicted in a legible size as possible.

Point 3. Figure 7 and 8 charts x axis legend is too complicated: is better to use the same of the western blot images, that id simpler, moreover this will render more homogeneous the entire figure.

Response 3. Thank you for pointing this out. As you suggested, we rewrite the x axis legend in the Figure 7-8 and checked all ID in figures are homogeneous.

Point 4. More specific issue could be raised for figure 3. This figure is not extensively commented in the text for instance stating differences between data for normal diet and for HFD. Moreover the charts are not clear because negative control is located at the center of values (I would suggest to reconsider output format of this charts). Moreover in H&E images should scale bar.

Response 4. Thank you for pointing this out. As you suggested, this figure is commented in Section 2.5. LMF-HSFx attenuates hepatic lipotoxicity and modulates adipogenesis in mice fed with high-fat diet, in line 163-185. Figure 3 now is moved to figure 6, the charts are rewrite and depicted in a legible size as possible. The scale bar in H&E image was added.

Point 5. Generally all abbreviations should be explained in the text the first time of mention.

Response 5. Thank you for pointing this out. As you suggested to check all abbreviations in manuscript and figures and proofread them.

Reviewer 2 Report

Ping-Hsiao Shih et al evaluated effects of Fucoidan and Fucoxanthin Atten on Hepatic Steatosis and Inflammation of NAFLD. After close evaluation of manuscript I would suggest revision according to the next points:

  1. Please consider to revise Keywords (Fucoidan; Fucoxanthin; hepatic steatosis are exists in te title).
  2. In Introduction: I would suggest to support the phrase "Fucoidan... is extensively investigated for its different biological activities on anti- inflammatory, anti-aggregation and anti-oxidative effects" with more recent publications (see https://doi.org/10.1016/j.carbpol.2021.117692; https://doi.org/10.3390/md18120601; https://doi.org/10.3390/md18050275).
  3. In Section 2.1: I would suggest to not mix in vivo and clinical results. The first phrase is confucing: "The baseline characteristics of both groups were similar..." Which groups you mean (animals, humans?). Next sentence "...patients with LMF-HSFx or placebo treatment..." should be clarified. What was a rate of LMF-HSFx, dose, schedule of treatment, what was a placebo?
  4. Fig 1: "70 were screened for eligibility" - please update 70 who?
  5.   The legend for figure 3 is not informative. Rate between LMF and HSFx, dose, OF-HF?
  6. Section 2.3" again mix of clinical and in vivo results.
  7. The discussion represents the mix of in vivo, in vitro and clinical data. It is not clear, which results are discussed: "...LMF-HSFx may enhance the leptin and adiponectin expression in adipocytes, and decreases insulin resistance..." in vitro? in vivo? in humans? "...LMF-HSFx suppresses lipotoxicity-induced liver injury...." in vitro? in vivo? in humans?
  8. Authors start discussion with Fig.5, then jump to Fig.7, then refer fig. 2,3, 7,8, etc. All looks very chaotic.
  9. In Sction 4.1: no information on age of patients (which enter the study after screening). It looks so that patients with NAFLD were treated with MF-HSFx capsules only during 6 months? The rationality f dose selection is not provided.
  10. In Sction 4.1:The LMF as well as HSFx should be characterized in details: molecular weight, sulfates, monosaccharides in LMW, concentration of fucoxanthin, the concertation of stabilizers for fucoxanthin and type of stabilizing agent should be mentioned. What was used as placebo?
  11. Section 4.2- rationality of doses selection for animals? How food consumption was recorded for each animal? Were animals of one group maintained in one cage? The analytical procedures should be described in more detaisl: which reagents were used?
  12. Section 4.4: concentrations of LMF-HSFx?
  13. Too many data in one manuscript. Two separate papers (one with in vivo/ in votro) results, other with clinical results) will be more suitable and clear.

Author Response

Response to Reviewer 2 comment

Point 1. Please consider to revise Keywords (Fucoidan; Fucoxanthin; hepatic steatosis are exists in te title). 

Response 1: Thank you for pointing this out. As you suggested to remove the Keywords, Fucoidan, Fucoxanthin and hepatic steatosis; to add the Keywords, randomized controlled trial; lipid metabolism; lipotoxicity; liver fibrosis.

Point 2. In Introduction: I would suggest to support the phrase "Fucoidan... is extensively investigated for its different biological activities on anti- inflammatory, anti-aggregation and anti-oxidative effects" with more recent publications (see https://doi.org/10.1016/j.carbpol.2021.117692; https://doi.org/10.3390/md18120601; https://doi.org/10.3390/md18050275).

Response 2. Thank you for your suggestions. We added the recent publications as references 5-7 in Introduction.

Point 3. In Section 2.1: I would suggest to not mix in vivo and clinical results. The first phrase is confucing: "The baseline characteristics of both groups were similar..." Which groups you mean (animals, humans?). Next sentence "...patients with LMF-HSFx or placebo treatment..." should be clarified. What was a rate of LMF-HSFx, dose, schedule of treatment, what was a placebo?

Response 3.1 Thank you for pointing this out. We reorganized the result sessions. Now, the session 2.1-2.4 and figure 1-5 are clinical trial data; session 2.5 and figure 6 are mice fed with high-fat diet data; session 2.6-7 and figure 7-8 are HepaRG cells data. We also add the model (clinical trial, mice fed with high-fat diet or HepaRG cells) in result session titles.

Point 3.2. The first phrase is confucing: "The baseline characteristics of both groups were similar..." Which groups you mean (animals, humans?).

Response 3.2 They are human groups, we have rewritten the sentences in line 78-79 as “Forty-two patients were equally randomized into LMF-HSFx and placebo group. The baseline characteristics of both groups were similar...”

 Point 3.3. Next sentence "...patients with LMF-HSFx or placebo treatment..." should be clarified. What was a rate of LMF-HSFx, dose, schedule of treatment, what was a placebo?

Response 3.2 Thank you for pointing this out. We now rewrote the sentences in line 85-87 as “we investigated the serum biochemical parameters in NAFLD patients with 24-week oral LMF-HSFx (825 mg LMF fucoidan plus 825 mg HSFx twice daily) or placebo (1650 mg cellulose powder twice daily) (Figure 1).”

Point 4. Fig 1: "70 were screened for eligibility" - please update 70 who?

Response 4. Thank you for pointing this out. We now rewrote the sentence as ”70 patients with NAFLD were screened for eligibility” in Figure 1

Point 5. The legend for figure 3 is not informative. Rate between LMF and HSFx, dose, OF-HF?

Response 5.1 Thank you for pointing this out. As you suggested, this figure is commented in Section 2.5. LMF-HSFx attenuates hepatic lipotoxicity and modulates adipogenesis in mice fed with high-fat diet, in line 163-185. Figure 3 now is moved to figure 6. We also rewrote the figure legend.

Point 5.2 Rate between LMF and HSFx, dose, OF-HF?

Response 5.2 Thank you for pointing this out. We used LMF plus HSFx as treatment in this study. The rate and dose of LMF-HSFx are 400 mg/kg bw/day for mice in Figure 6A and 6B; The rate and dose of LMF-HSFx are 0, 200 or 400 mg/kg bw/day in Figure 6C-6F. We apologize for the abbreviation mistake “OF-HF” in Figure 6. We now rewrote the abbreviation as “LMF-HSFx (mg/kg bw)”.

Point 6 Section 2.3" again mix of clinical and in vivo results.

Response 6 Thank you for pointing this out. We reorganized the result sessions. Now, the session 2.1-2.4 and figure 1-5 are clinical trial data; session 2.5 and figure 6 are mice fed with high-fat diet data; session 2.6-7 and figure 7-8 are HepaRG cells data.

Point 7 The discussion represents the mix of in vivo, in vitro and clinical data. It is not clear, which results are discussed: "...LMF-HSFx may enhance the leptin and adiponectin expression in adipocytes, and decreases insulin resistance..." in vitro? in vivo? in humans? "...LMF-HSFx suppresses lipotoxicity-induced liver injury...." in vitro? in vivo? in humans?

Response 7 Thank you for pointing this out. We now rewrote the sentences in discussion, line 261-275: “Here, in clinical trial, we provide evidence that LMF-HSFx has the potential against lipotoxicity (Figure 2), hepatic steatosis (Figure 3), inflammation (Figure 4) and insulin resistance (Figure 5) in NAFLD patientsSimilar results were also confirmed in HFD-induced-NAFLD mice…… Briefly, in vivo, LMF-HSFx may enhance the leptin and adiponectin expression…”

Point 8 Authors start discussion with Fig.5, then jump to Fig.7, then refer fig. 2,3, 7,8, etc. All looks very chaotic.

Response 8 Thank you for pointing this out. We now rewrote the sentences in discussion, line 261-274: “Here, in clinical trial, we provide evidence that LMF-HSFx has the potential against lipotoxicity (Figure 2), hepatic steatosis (Figure 3), inflammation (Figure 4) and insulin resistance (Figure 5) in NAFLD patients…LMF-HSFx also significantly decreased the HFD-induced hepatic lipotoxicity in mice, such as lipid droplet sizes, blood glucose level, triglyceride level, AST and ALT (Figure 6). Following, LMF-HSFx also significantly up-regulated the adiponectin expression genes, adipoq and adig, and leptin expression gene lep in adipose tissue and pro-brown adipose tissue in HFD-fed mice (Table 1)…Next, adiponectin triggers the Adiponectin-AdipoR1/2 and SIRI-PGC-1 pathways in the hepatocyte cell (Figure 7)...” All figure numbers are shown in sequence as possible.

Point 9 In Sction 4.1: no information on age of patients (which enter the study after screening). It looks so that patients with NAFLD were treated with MF-HSFx capsules only during 6 months? The rationality f dose selection is not provided.

Response 9.1 Thank you for pointing this out. We now rewrote the sentences in line 360-361 “…forty-two patients, age (years, 55±12.5 vs 59±10.5, p=0.23), were enrolled into the study and randomized into two groups...” The detail of baseline characters was described in Supplementary table 1. Yes, in LMF-HSFx group, they only took LMF-HSFx as treatment reagent during 6 months.

Point 9.2 The rationality of dose selection is not provided.

Response 9.2 In previous animal study (reference doi: 10.3390/md15040113), the rationality dose of LMF-HSFx was 600 mg/kg/day. In order to use this drug to human, we determined the human dose according to the animal dose, so, it was 3 g/day for a 60 kg human (reference doi: 10.4103/0976-0105.177703). In this clinical trial, LMF-HSFx dose in each capsule was 0.55 g, therefore, approximate 6 capsules of LMF-HSFx were applied as the daily dose.

Point 10 In Sction 4.1: The LMF as well as HSFx should be characterized in details: molecular weight, sulfates, monosaccharides in LMW, concentration of fucoxanthin, the concertation of stabilizers for fucoxanthin and type of stabilizing agent should be mentioned. What was used as placebo?

Response 10.1 Thank you for pointing this out. We have added the sentences in line 377-382, “The LMF-HSFx capsules in this trial were derived from Sargassum hemiphyllum and prepared by HiQ Marine Biotech, Taipei, Taiwan. The average molecular weight, monosaccharide fucose content, and sulfate content of LMF used in this study were 0.8 KDa (92.1%), 210.9 ± 3.3 mmol/g, and 38.9% ± 0.4% (w/w), respectively. HSFx is a mixture containing about 10% of fucoxanthin that is coated directly with polysaccharides of its own.” And reference 34 and 35. https://doi.org/10.3390/md16100392; https://doi: 10.3402/fnr.v60.32033

Point 10.2 What was used as placebo?

Response 10.2 Cellulose powder was used as placebo. We now rewrote the sentences in line 362-365, ”Subjects took 3 capsules of LMF-HSFx…, or placebo (3 capsules of 550 mg/capsule cellulose powder) in the control group.”

Point 11 Section 4.2- rationality of doses selection for animals? How food consumption was recorded for each animal? Were animals of one group maintained in one cage? The analytical procedures should be described in more detaisl: which reagents were used?

Response 11 In previous animal study (reference doi: 10.3390/md15040113), rationality dose of LMF was 300 mg/kg/day. In this paper, the body weight for each mouse and food consumption from each cage were recorded and average every day. We prepare two cages for each group, so there are 3 or 4 mice in per cage. We have added the words “3-4 mice per cage” in line 391-392. In Section 4.2, for the blood glucose level, triglyceride level, AST and ALT concertation were analyzed by automated biochemical analyzer directly from the serum of blood.  No reagents were required for the serum biochemistry study. For Hematoxylin and eosin stain, the reagents were hematoxylin and eosin, we have written in line 401-404 “Right lobe of liver was fixed with 4% buffered paraformaldehyde and embedded in paraffin for further hematoxylin and eosin stain…”

Point 12 Section 4.4: concentrations of LMF-HSFx?

Response 12 Thank you for pointing this out. We now rewrite the sentence in line 423, “At the day of treatment, the medium…in the presence or absence of 25 mg/ml LMF-HSFx…”

Point 13 Too many data in one manuscript. Two separate papers (one with in vivo/ in votro) results, other with clinical results) will be more suitable and clear.

Response 13 Thank you for your kind suggestion. We try clarify the manscript and hope it is suitable for further processing.

Round 2

Reviewer 2 Report

The clarity of presentation was significantly improved after revision. However, some points needs additional revision. As soon as authors decide to provide all information in one paper I suggest to improve and extend discussion part.

  1.  One important point should be addressed: pharmacokinetic. There are some important publications related to pharmacokinetic of marine derived drugs. Please discuss tissue distribution of  fucoidan and fucoxanthin and other pharmacokinetic parameters.
  2. I would suggest to compare effects observed by authors with effects described for fucoidan and fucoxanthin in literature.
  3. The LMF-HSFx capsules comprise fixed combination of two compounds. What about synergy? Please discuss this as well.
  4. Please provide the characterization of the LMF-HSFx capsules in separate subsection.
  5. Please revise the conclusion according to new updated discussion.

Author Response

Response to Reviewer 2 comment

Point 1. One important point should be addressed: pharmacokinetic. There are some important publications related to pharmacokinetic of marine derived drugs. Please discuss tissue distribution of fucoidan and fucoxanthin and other pharmacokinetic parameters. 

Response 1. Thank you for your suggestions. We added the conclusion in Discussion section, line 299-305. “The pharmacokinetic and tissue distribution of active compounds is essential for drug development process…Therefore,..” and added two publications as references. doi: 10.3390/md16040132 doi: 10.1017/S0007114508199007.

Point 2. I would suggest to compare effects observed by authors with effects described for fucoidan and fucoxanthin in literature.

Response 2. Thank you for your suggestions. We added the conclusions in Discussion section, line 295-298 “In db/db mice, LMF alone…The anti-obesity effects of HSFx…”;line 305-312, “…LMF was more effective than HSFx…urinary sugar and reduce in inflammatory adipocytokines were significantly observed in LMF-HSFx group, but not in HSFx or LMF alone…”; line 320-325, “In rat, fucoidan alone was reported to... Similar effects were observed for Fucoxanthin alone...”; line 342-344,” Similar pathway modulation for insulin resistance…”

Point 3. The LMF-HSFx capsules comprise fixed combination of two compounds. What about synergy? Please discuss this as well.

Response 3 Thank you for pointing this out. We added this conclusion in Discussion section, line 310-312, “Interestingly, the regulation efficacies of LMF-HSFx were better than HSFx or LMF alone on urinary sugar decreasing, glucose and lipid metabolism in white adipose tissue, indicating a synergistic effect of LMF and HSFx”

Point 4. Please provide the characterization of the LMF-HSFx capsules in separate subsection.

Response 4 Thank you for pointing this out. As you suggested, the characterization of the LMF-HSFx capsules was moved to new Section 4.1. Materials.

 Point 5. Please revise the conclusion according to new updated discussion.

Response 5 Thank you for your suggestion. The new conclusions were added in paragraph 2-5 in discussion. In the finally conclusion, we added the sentence in line 365-366, “In addition, combination of LMF and HSFx also shows a synergistic modulation on glucose and lipid metabolism.”

Round 3

Reviewer 2 Report

The manuscript was revised and suitable for publication.

Author Response

Thank you for your kind suggestion